# Identification of potential inflammation markers for outgrowth of cow's milk allergy

Diana M. Hendrickx[ID][1]*, Mengyichen Long[1], PRESTO study team[¶], Harm Wopereis[2], Renate G. van der Molen[3☯], Clara Belzer[ID][1☯]*

1 Laboratory of Microbiology, Wageningen University, Wageningen, The Netherlands, 2 Danone Nutricia Research, Utrecht, The Netherlands, 3 Department of Laboratory Medicine, Laboratory of Medical Immunology, Radboudumc, Nijmegen, The Netherlands

¶ Membership of the PRESTO study team is provided after the Acknowledgments.
☯ These authors contributed equally to this work.
* diana.hendrickx@wur.nl (DMH); clara.belzer@wur.nl (CB)

## Abstract

Immunoglobulin E (IgE)-mediated cow's milk allergy (CMA) is an immune-mediated reaction to cow's milk (CM). Non-IgE-mediated CMA resolves in most children in the first years of life, whereas IgE-mediated CMA outgrowth is often later or not at all. The exact mechanisms underlying resolution of IgE-mediated CMA are not fully understood. We aim to gain insight in the immunological mechanisms underlying resolution of IgE-mediated CMA by analyzing unique saliva samples of allergic infants using the Olink® Target 96 Inflammation panel. Twenty-four children who outgrew their CMA after 12 months were compared to 15 with persistent CMA. Persistent CMA was accompanied by an increase in interleukin-15 receptor subunit alpha in the first 6 months, followed by a decrease, hinting towards an initial increased T cell response. At the same time caspase-8 was increased and interleukin-7 was decreased in persistent CMA. For CMA resolution, we found elevated levels of delta and notch-like epidermal growth factor-related receptor. Furthermore, adenosine deaminase (ADA) increased significantly between 0 and 12 months in resolved CMA, but not in persistent CMA. KEGG pathway analysis suggests mainly the TNF signaling pathway to be important in the resolution of CM allergy. Our findings show that Olink® Target 96 Inflammation panel analysis of saliva samples can reveal potential immunological markers and mechanisms involved in CMA resolution.

## Introduction

Cow's milk allergy (CMA) occurs when the immune system overreacts to proteins found in cow's milk (CM). Immune-mediated reactions to CM are categorized as immunoglobulin E (IgE)-mediated, non-IgE-mediated or mixed [1]. Most infants that did not outgrow their allergy within 1 year after diagnosis have IgE-mediated CMA [2]. IgE-mediated CMA occurs when after exposure to CM proteins via the skin,

**Data availability statement:** The Olink Target 96 Inflammation panel data and R code have been deposited in Gitlab (https://git.wur.nl/afsg-microbiology/publication-supplementary-materials/2024-hendrickx-et-al-early-fit-presto-immune-study). The clinical study data supporting the findings of this study are available on request from Danone Research & Innovation (Uppsalalaan 12, 3584 CT Utrecht, The Netherlands, https://www.danoneresearch.com/ ); understanding that there could be reasonable caveats for such requests. Researchers that meet the criteria for access to confidential clinical study data must be compliant with the Danone Research & Innovation Clinical Trial Dataset Sharing policy. Danone conducts clinical studies according to the ICH-GCP guidelines, the Declaration of Helsinki, and the WHO code. In addition, the quality management system of Danone for clinical research has been ISO 9001 certified since 2007 and has been recertified every three years. Certified compliance with ICH-GCP standards provides public assurance that the rights, safety, and wellbeing of trial subjects are protected, and that clinical-trial data are scientifically credible. Study data will be retained by Danone Research & Innovation for a period of at least 25 years for the purpose of verification of data and results, scientific research compatible with data protection laws (e.g. GDPR Article 89) and to comply with other relevant legal requirements.

**Funding:** CB, RGvdM and DMH were funded by the Dutch Research Council (NWO) and Danone Nutricia Research (grant no.16490, received by CB). The project was part of the partnership programme between NWO-TTW and Danone-Nutricia Research (https://www.nwo.nl/en/researchprogrammes/partnership/partnership-programmas/danone-nutricia-research). The Dutch Research Council (NWO) played no role in the study design, data collection and analysis, the decision to publish and the preparation of the manuscript. Danone Nutricia Research provided the clinical data and saliva samples for this study. HW is an employee of Danone-Nutricia Research. HW reviewed the manuscript, provided comments and approved the article.

**Competing interests:** The authors of this manuscript have the following competing interests: H.W. is an employee of Danone-Nutricia

gastrointestinal tract or respiratory system, people develop IgE antibodies against CM. During the sensitization phase, antigens are processed by antigen presenting cells and presented to naïve T cells, which further differentiate into T helper (Th) 2 cells. The latter then interact with B cells, resulting in an immunoglobulin class switch and the production of CM protein-specific IgE antibodies, which bind to mast cells. During the effector phase, re-exposure to the allergen causes cross-linking of the mast cell-bound IgE and thus induce the degranulation of mast cells, and the release of inflammatory mediators such as histamine, causing allergic reactions within a few hours after exposure [3].

Resolution of IgE-mediated CMA is in most cases achieved within the first 3 years of life [4]. The exact mechanisms are not known, but there is an agreement that individuals reaching higher CM-specific IgE levels have a lower probability to outgrow their CMA [5]. Furthermore atopic dermatitis, other food allergy and respiratory allergy have been identified as predictive for persistent CMA [5]. Infant formula supplemented with a probiotic, prebiotic or synbiotic blend have been proposed to accelerate resolution of IgE-mediated CMA [6].

It has been reported that CM-specific IgE, CM-allergen-specific IgE, basophil activation, CM-allergen-specific T cells and gut microbiota show some evidence for the resolution of CMA [7]. However, these markers do not provide a clear answer to the question of which immunological features are important in resolution of CMA.

Previous studies have shown that T cells from patients with CMA differ in their Th1/Th2 balance compared to healthy individuals by producing significantly lower levels of Th1 cytokines and significantly higher levels of Th2 cytokines [8,9]. In people who outgrew their CMA, a shift to a Th1 cell response, with increased Th1 cytokines and decreased Th2 cytokines, was observed [10].

In this study, we aimed to get a better understanding of the mechanisms underlying resolution of IgE-mediated CMA by analyzing unique saliva samples from infants with IgE-mediated CMA using the Olink® Target 96 Inflammation panel, enabling us to measure 92 proteins related to inflammation, including Th1 and Th2 cytokines. Measuring these inflammation factors allows us to discriminate between different T cell responses. We studied the changes of these proteins in children who outgrew CMA after 12 months compared to those with persistent CMA. Moreover, we compared Olink® inflammation panel proteins between infants who received a standard amino acid-based formula (AAF) and those who received an amino acid-based formula supplemented with a synbiotic blend (AAF-syn).

## Methods

### Sample collection and experimental design

In our study, which was part of the EARLYFIT project, 39 infants who participated in the PRESTO clinical trial (NTR3725) [11] were selected based on sample and data availability as described elsewhere [12], a flowchart of the subject selection process is provided in S1 Fig in S1 File. The infants selected for our study originated from 10 sites in 6 countries (Southampton General Hospital – United Kingdom, Charité

Research. This study was part of the EARLYFIT project, which is part of a partnership programme between NWO-TTW and Danone-Nutricia Research. The research of DMH, CB and RGvdM was funded by the Dutch Research Council (NWO) and Danone Nutricia Research . This does not alter our adherence to PLOS ONE policies on sharing data and materials.

Hospital Berlin – Germany, St.-Marien Hospital – Germany, Ruhr-Universität Bochum im St. Josef-Hospital – Germany, University Hospital Verona – Italy, KK Women's & Children's Hospital – Singapore, King Chulalongkorn Memorial Hospital – Thailand, Ramathibodi Hospital – Thailand, Prince of Songkla Hospital – Thailand, Texas Children's Hospital – United States of America).

In the PRESTO study, infants diagnosed with IgE-mediated CMA received specialized infant formula for the dietary management of their allergy (AAF or AAF-syn). The synbiotic blend consisted of probiotic *Bifidobacterium breve* M-16V and prebiotic oligosaccharides (oligofructose, inulin)(dosage described by Chatchatee et al [11]). For both formula, the acquisition of tolerance to CM was similar and in line with natural outgrowth of CMA [11]. As described earlier, infants for the PRESTO study were recruited between August 7, 2013 and February 6, 2017 [11]. For the infants selected for this study, the recruitment period was August 7, 2013 until January 30, 2017. The recruitment period differed between study sites. At Southampton General Hospital, the infants for PRESTO were recruited between June 12, 2014 and May 23, 2016; the selected infant for our study was recruited on July 27, 2015. At Charité Hospital Berlin, the infants for PRESTO were recruited between January 9, 2014 and December 6, 2016; for the selected infants in our study, the start and end date of recruitment was the same. At St.-Marien Hospital, the infants for PRESTO were recruited between September 20, 2013 and September 29, 2016; the selected infants for our study were recruited between September 20, 2013 and October 22, 2013. At Ruhr-Universität Bochum im St. Josef-Hospital, the infants for PRESTO were recruited between May 2, 2014 and March 10, 2015; the selected infant for our study was recruited on March 10, 2015. At University Hospital Verona, the infants for PRESTO were recruited between March 27, 2014 and May 27, 2014; the selected infant for our study was recruited on March 27, 2014. At KK Women's & Children's Hospital, the infants for PRESTO were recruited between October 8, 2013 and March 1, 2016; the selected infants for our study were recruited between November 19, 2013 and March 1, 2016. At King Chulalongkorn Memorial Hospital, the infants for PRESTO were recruited between August 15, 2013 and January 30, 2017; the selected infants for our study were recruited between February 26, 2014 and January 30, 2017. At Ramathibodi Hospital, the infants for PRESTO were recruited between September 26, 2013 and December 29, 2016; the selected infants for our study were recruited between September 26, 2013 and June 19, 2015. At Prince of Songkla Hospital, the infants for PRESTO were recruited between August 7, 2013 and October 27, 2016; the selected infants for our study were recruited between August 7, 2013 and December 30, 2015. At Texas Children's Hospital, the infants for PRESTO were recruited between June 27, 2014 and July 20, 2016; the selected infants for our study were recruited between July 18, 2014 and December 1, 2014.

IgE-mediated CMA was determined as described elsewhere [11]. For more detail, see Section 1 in S1 File. Clinical characteristics at enrollment are presented in S1 Table in S1 File. Resolution of CMA after 12 months of the start of the study was demonstrated as previously described [11] as provided in Section 2 in S1 File.

Saliva samples from 39 infants were collected in the clinic at least 1 hour after feeding using the SalivaBio Children's Swab method (Salimetrics, Carlsbad, USA). Samples were immediately frozen at −80°C and transported on dry-ice (solid CO2) to Danone Nutricia Research. Samples were aliquoted, transported on dry-ice to Radboudumc Nijmegen and stored at −80°C until analysis.

For the 39 infants, samples were collected at 3 visits (baseline – after CMA diagnosis, 6 and 12 months after entering the study), with the exception of one infant with persistent CMA who received AAF-syn and for whom only samples at baseline and visit 6 months were collected.

At the start of the study, all infants were aged 3–13 months (mean age ca 9 months). Twelve months after entering the study, twenty-four infants (10 AAF, 14 AAF-syn) resolved their CMA, while 15 (6 AAF, 9 AAF-syn) were still allergic to CM. Baseline characteristics per allergy group are presented in S2 Table in S1 File. Furthermore, IgE-specific α-lactalbumin, β-lactoglobulin and casein are significantly lower in outgrown CMA than in persistent CMA (S3 Table in S1 File). More detailed clinical characteristics per allergy group and per treatment group are provided elsewhere [12,13].

## Ethical approval

Ethical approval was obtained as described elsewhere [11]. This multicenter study was performed according to the World Medical Association (WMA) Declaration of Helsinki and the International Conference on Harmonization guidelines for Good Clinical Practice [11]. S4 Table in S1 file provides an overview of all ethics committees, institutional review boards and regulatory authorities that approved this study. Written informed consent for the collection and analysis of the data was obtained from the parents of all infants included in this study [11].

## Preprocessing and quality control

Ninety-two proteins related to inflammation and immune response were measured with the Olink® Target 96 Inflammation panel (Olink Proteomics, Uppsala, Sweden). An overview is provided on the Olink® website (https://olink.com/products/olink-target-96).

Olink® measures many markers simultaneously, requires only a small sample volume, has a wide measuring range and is very sensitive. This makes the method very suitable for saliva samples for which concentrations of immune markers are not very high.

The aliquoted saliva samples were directly added to the plates, and were randomly distributed over two plates to avoid batch effects. The Olink® Target 96 Inflammation panel uses the proximity extension assay (PEA) [14] protocol with quantitative PCR (qPCR) readout. For more detail, see Section 3 in S1 File.

Threshold cycle (Ct) values from the qPCR were transformed to Normalized Protein Expression (NPX) values and normalized between plates as described in Section 4 in S1 File. Quality control (QC) was performed as described in Section 5 in S1 File.

Proteins below the limit of detection (LOD) in more than 20% of the samples were removed from the data set and not considered for further analysis.

The results of the quality control and filtering are provided in Section 6 in S1 File and S2 Fig. Fifty-eight proteins, listed in S5 Table in S1 File were included in the statistical analysis.

## Statistical analysis

Multivariate analysis with Repeated Measures ANOVA–Simultaneous Component Analysis Plus (RM-ASCA+) [15] was performed using R version 4.2.1 [16] and the R package ALASCA [17] to study changes in inflammation markers between visits and differences between allergy groups (resolution of CMA at 12 months (yes/no)) or treatment groups (AAF/AAF-syn). In RM-ASCA+, effects of experimental variables are estimated with linear mixed models (LMM), and used as input for principal component analysis (PCA) [15].

Linear mixed models (LMM) analysis was performed using the R package OlinkAnalyze version 3.3.1 [18] to determine differences between allergy groups (resolution of CMA at 12 months (yes/no)) or treatment groups (AAF/AAF-syn) within visits, and between visits within allergy or treatment groups. For more details, see Section 7 in S1 File.

To identify (KEGG) [19] pathways which show statistically significant, concordant differences between allergy or treatment groups within visits, and between visits within allergy or treatment groups, enrichment analysis was performed in ConsensusPathDB [20]. For more details, see Section 8 in S1 File.

As inflammation markers may be influenced by infections, we also compared the number of infections between 6-month visit intervals within allergy and treatment groups by using a paired two-sided Mann Whitney U-test. The number of infections were also compared between allergy and treatment groups within 6-month visit intervals by means of a two-sided Mann Whitney U-test.

## Results

### Development of immune factors over visits within allergy groups

Analysis of the combined effect of visit and allergy group (Fig 1A) shows increasing PC1 scores in both allergy groups, with consistently higher PC1 scores for the infants with resolved CMA. The overlap between the confidence intervals in

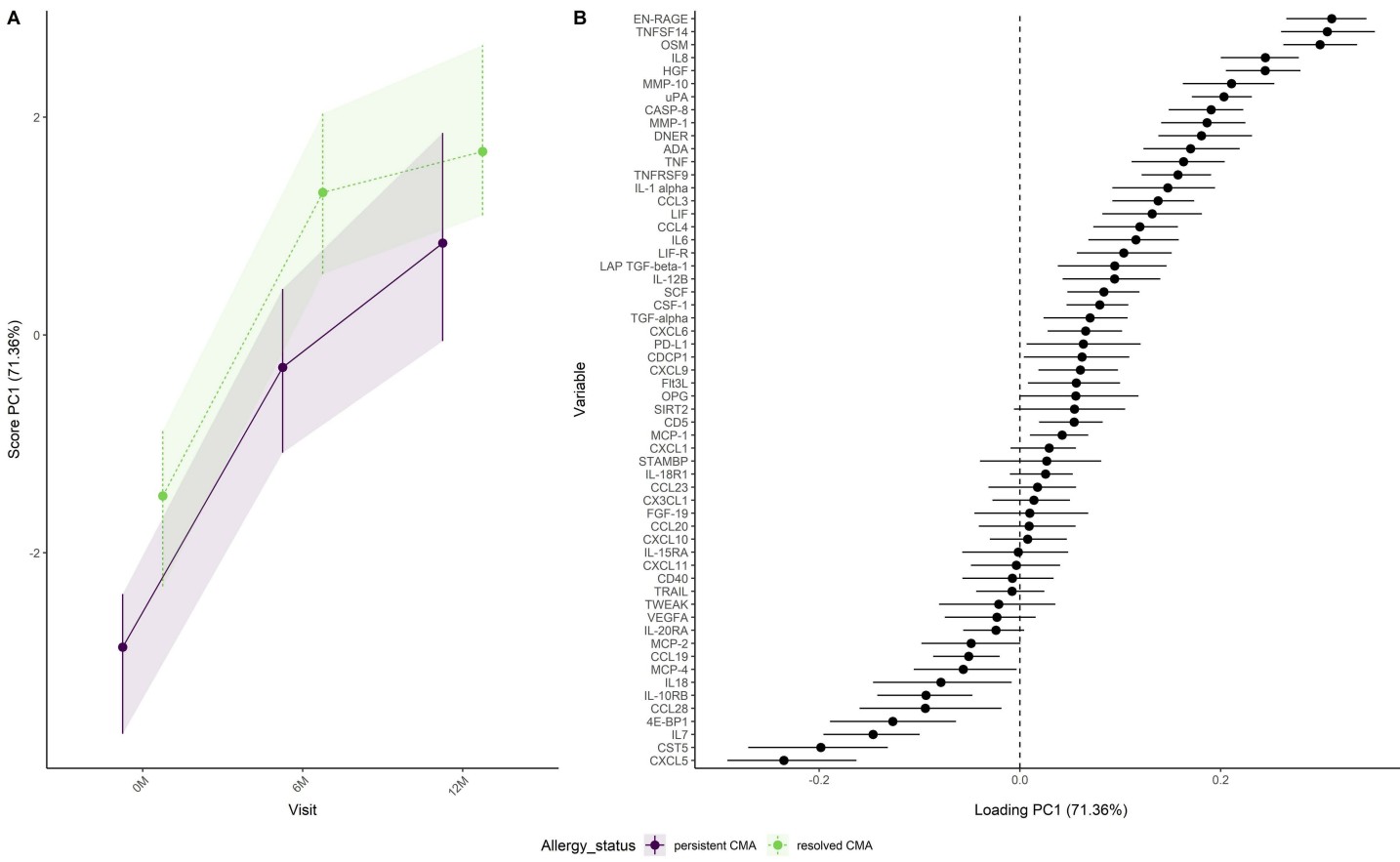

**Fig 1. Development of immune factors over visits in the persistent and resolved CMA group as (A) scores and (B) loadings.** A positive loading means that the NPX value of the immune factor increases when the scores increase. A negative loading means that the NPX value of the immune factor decreases when the scores increase.

Fig 1A suggests larger differences in immune factors between visits than between allergy groups. Immune factors, like EN-RAGE, TNFSF14 and OSM had a higher NPX value in the group with resolved CMA compared to the group with persistent CMA (Fig 1B). Lower NPX values in the group with resolved CMA compared to persistent CMA were obtained for immune factors like CXCL5, CST5 and IL7 (Fig 1B).

Fig 2A adjusts for trends over visits in the group with persistent CMA by using this group as a reference. The figure shows how the immune factors in the group with resolved CMA differ from the persistent allergy group. We observed that the largest difference between the allergy groups occurs at visit 6M (Fig 2A). Scores were lower for the group with resolved CMA than for the group with persistent CMA (Fig 2A). After adjusting for trends over visits in the group with persistent CMA, infants with resolved CMA showed higher NPX values for immune factors like EN-RAGE, MMP-10 and PD-L1, while lower NPX values were observed for immune factors like CXCL5, IL-15RA and IL-10RB (Fig 2B).

## Significant differences between visits within allergy groups

S6 Table shows the results of the LMM analysis and subsequent post-hoc test to determine significant differences between visits within allergy groups. In persistent CMA, interleukin-15 receptor subunit alpha (IL-15RA) showed an increase between baseline and 6 months, followed by a decrease between 6 and 12 months (see Table 1, S6 Table, and

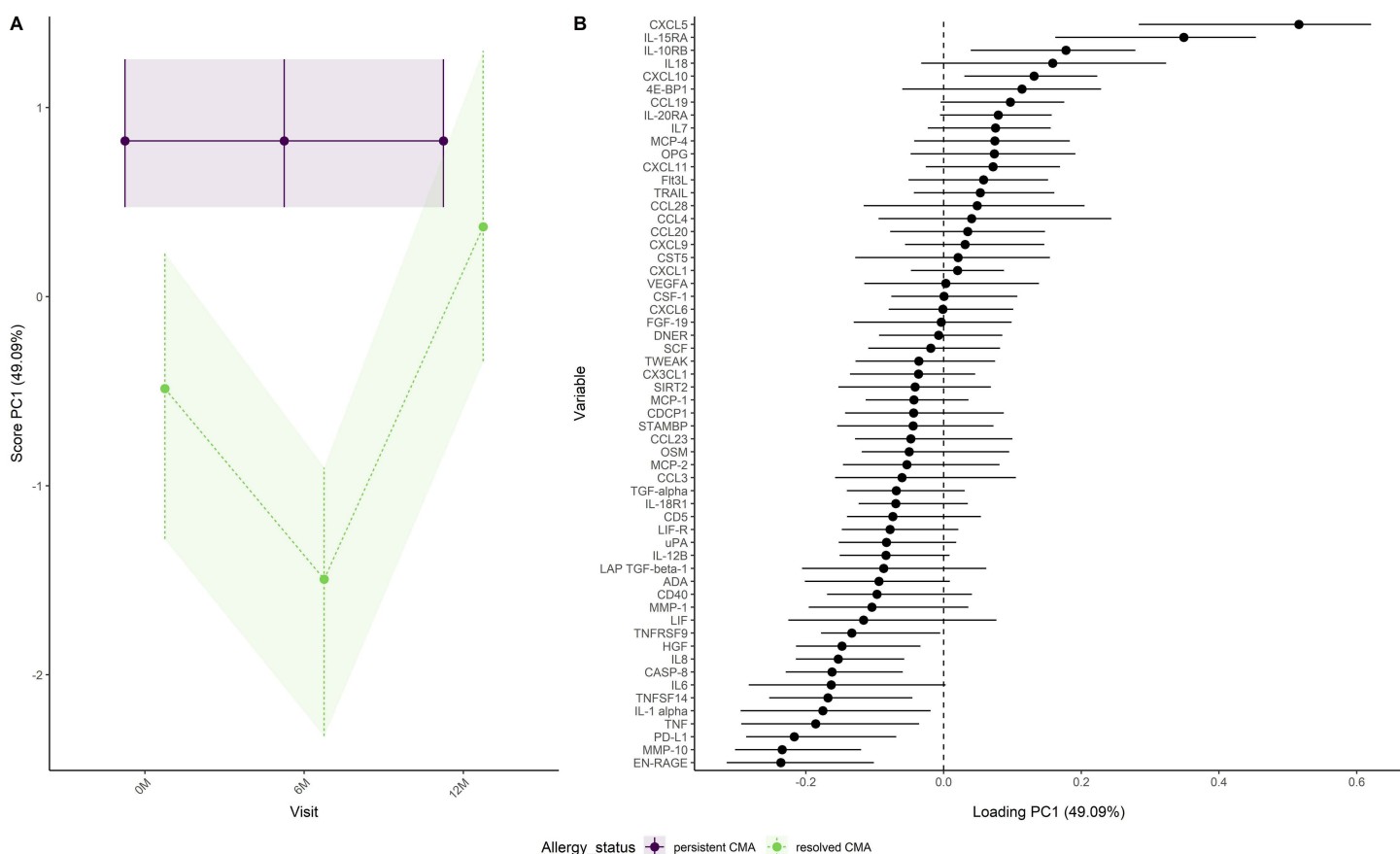

**Fig 2. Development of immune factors over visits in the persistent and resolved CMA group as (A) scores and (B) loadings when using the persistent CMA group as a reference.** A positive loading means that the NPX value of the immune factor increases when the scores increase. A negative loading means that the NPX value of the immune factor decreases when the scores increase.

boxplots in S3 Fig in S1 File). In contrast, no significant changes in IL-15RA were observed in the saliva of infants who did outgrow their CMA (Table 1, S6 Table, S3 Fig in S1 File). For oncostatin-M (OSM), cystatin D (CST5) and hepatocyte growth factor (HGF) similar changes between visits were observed in both allergy groups (Table 1, S6 Table, S3 Fig in S1 File). Levels of tumor necrosis factor ligand superfamily member 14 (TNFSF14) and protein S100-A12 (EN-RAGE) increased over visits in both allergy groups (Table 1, S6 Table, S3 Fig in S1 File). However, in resolved CMA the increase

**Table 1. Proteins with at least one significant difference (adjusted p-value ≤ 0.05) between visits within allergy groups as determined by linear mixed models and post hoc analysis.**

| Protein | Persistent CMA at visit 12M Number of samples: 0M: n = 15, 6M: n = 15; 12M: n = 14 | | | | | |
| --- | --- | --- | --- | --- | --- | --- |
| | Estimate and 95% CI for the difference in mean NPX | | | Adjusted p-value | | |
| | 0M to 6M | 6M to 12M | 0M to 12M | 0M to 6M | 6M to 12M | 0M to 12M |
| IL-15RA | **+0.77 [+0.19, +1.35]** | **−0.84 [−1.42,-0.26]** | −0.07 [−0.65, +0.51] | **0.0028** | **0.0009** | 0.9993 |
| OSM | **+2.05 [+0.38, +3.73]** | +0.53 [−1.15, +2.21] | **+2.58 [+0.99, +3.68]** | **0.0076** | 0.9401 | **0.0003** |
| CST5 | **−0.52 [−1.03,-0.02]** | −0.13 [−0.64, +0.38] | **−0.65 [−1.16,-0.14]** | **0.0400** | 0.9751 | **0.0044** |
| TNFSF14 | +1.21 [−0.23, +2.65] | +1.12 [−0.32, +2.56] | **+2.33 [+0.89, +3.77]** | 0.1510 | 0.2181 | **0.0001** |
| EN-RAGE | +1.98 [−0.04, +3.99] | +0.48 [−1.54, +2.49] | **+2.45 [+0.43, +4.47]** | 0.0587 | 0.9827 | **0.0084** |
| IL-1 alpha | +0.09 [−0.63, +0.81] | +0.35 [−0.37, +1.07] | +0.43 [−0.29, +1.15] | 0.9993 | 0.7180 | 0.4964 |
| HGF | +0.83 [−0.40, +2.06] | +0.56 [−0.67, +1.79] | **+1.39 [+0.16, +2.62]** | 0.3681 | 0.7713 | **0.0180** |
| IL-7 | −0.48 [−1.27, +0.30] | −0.40 [−1.19, +0.39] | **−0.88 [−1.67,-0.10]** | 0.4723 | 0.6741 | **0.0185** |
| CASP-8 | +1.30 [−0.20, +2.80] | +0.23 [−1.28, +1.73] | **+1.53 [+0.03, +3.03]** | 0.1272 | 0.9978 | **0.0438** |
| uPA | +0.64 [−0.60, +1.89] | +0.12 [−1.12, +1.34] | +0.76 [−0.48, +2.01] | 0.6573 | 0.9997 | 0.4739 |
| DNER | +0.20 [−0.07, +0.47] | +0.05 [−0.21, +0.32] | +0.25 [−0.01, +0.52] | 0.2636 | 0.9910 | 0.0741 |
| IL-8 | +0.88 [−0.32, +2.07] | +0.16 [−1.04, +1.35] | +1.04 [−0.16, +2.23] | 0.2754 | 0.9988 | 0.1269 |
| ADA | +0.46 [−0.61, +1.52] | +0.39 [−0.68, +1.45] | +0.85 [−0.22, +1.91] | 0.8054 | 0.8947 | 0.1978 |
| Protein | Resolved CMA at visit 12M Number of samples: 0M: n = 24, 6M: n = 24; 12M: n = 24 | | | | | |
| | Estimate and 95% CI for the difference in mean NPX | | | Adjusted p-value | | |
| | 0M to 6M | 6M to 12M | 0M to 12M | 0M to 6M | 6M to 12M | 0M to 12M |
| IL-15RA | +0.04 [−0.42, +0.50] | −0.12 [−0.59, +0.34] | +0.08 [−0.55, +0.38] | 0.9998 | 0.9700 | 0.9951 |
| OSM | **+1.67 [+0.35, +3.00]** | +0.66 [−0.68, +2.01] | **+2.34 [+0.99, +3.68]** | **0.0054** | 0.6696 | **< 0.0001** |
| CST5 | **−0.51 [−0.91,-0.11]** | −0.16 [−0.57, +0.25] | **−0.67 [−1.08,-0.26]** | **0.0048** | 0.8558 | **0.0001** |
| TNFSF14 | **+1.67 [+0.53, +2.81]** | +0.59 [−0.56, +1.74] | **+2.26 [+1.11, +3.41]** | **0.0007** | 0.6660 | **< 0.0001** |
| EN-RAGE | **+1.87 [+0.28, +3.47]** | −0.04 [−1.65, +1.58] | **+1.84 [+0.22, +3.45]** | **0.0120** | 1.0000 | **0.0164** |
| IL-1 alpha | **+0.63 [+0.06, +1.20]** | −0.14 [−0.72, +0.43] | +0.49 [−0.09, +1.07] | **0.0206** | 0.9774 | 0.1451 |
| HGF | +0.93 [−0.05, +1.90] | +0.40 [−0.59, +1.38] | **+1.32 [+0.34, +2.31]** | 0.0711 | 0.8459 | **0.0025** |
| IL-7 | −0.59 [−1.21, +0.03] | +0.11 [−0.52, +0.74] | −0.49 [−1.12, +0.14] | 0.0694 | 0.9962 | 0.2237 |
| CASP-8 | +1.05 [−0.14, +2.24] | −0.18 [−1.38, +1.02] | +0.87 [−0.33, +2.07] | 0.1140 | 0.9980 | 0.2901 |
| uPA | +0.82 [−0.17, +1.80] | +0.43 [−0.57, +1.42] | **+1.25 [+0.25, +2.24]** | 0.1589 | 0.8089 | **0.0061** |
| DNER | +0.14 [−0.07, +0.35] | +0.12 [−0.10, +0.33] | **0.25 [+0.04, +0.47]** | 0.3942 | 0.6082 | **0.0099** |
| IL-8 | +0.79 [−0.16, +1.73] | +0.35 [−0.60, +1.31] | **+1.14 [+0.18, +2.10]** | 0.1577 | 0.8894 | **0.0104** |
| ADA | +0.72 [−0.12, +1.56] | +0.22 [−0.63, +1.07] | **+0.94 [+0.09, +1.80]** | 0.1335 | 0.9733 | **0.0210** |

95% confidence intervals (CI) for the difference in mean NPX value and adjusted p-value (Benjamini-Hochberg). More detailed results are shown in S6 Table. Significant results are indicated in bold. +: increase; -: decrease.

Abbreviations: IL-15RA: interleukin-15 receptor subunit alpha; OSM: oncostatin-M; CST5: cystatin D; TNFSF14: tumor necrosis factor ligand superfamily member 14; EN-RAGE: protein S100-A12; IL-1 alpha: interleukin-1 alpha; HGF: hepatocyte growth factor; IL-7: interleukin-7; CASP-8: caspase-8; uPA: urokinase-type plasminogen activator; DNER: delta and notch-like epidermal growth factor-related receptor; IL-8: interleukin-8; ADA: adenosine deaminase.

was already significant between 0 and 6 months, while for persistent CMA significance was only observed after 12 months (Table 1, S6 Table, S3 Fig in S1 File). Interleukin-7 (IL-7) and caspase-8 (CASP-8) significantly decreased and increased respectively between 0 and 12 months in persistent CMA, but not in resolved CMA (Table 1, S6 Table, S3 Fig in S1 File). Urokinase-type plasminogen activator (uPA), delta and notch-like epidermal growth factor-related receptor (DNER), interleukin-8 (IL-8) and adenosine deaminase (ADA) increased significantly between 0 and 12 months in resolved CMA, but not in persistent CMA (Table 1, S6 Table, S3 Fig in S1 File). No significant differences between allergy groups were found within visits (S7 Table).

## Development of immune factors over visits within treatment groups

Analysis of the combined effect of visit and treatment group (Fig 3A) showed an increase followed by a slight decrease in PC1 score for the AAF group, while an increasing trend for the AAF-syn group was observed. The overlap between the confidence intervals in Fig 3A suggests larger differences in immune factors between visits than between treatment groups. PC1 scores were lower for the AAF-syn group than for the AAF group at baseline and visit 6 months, while at visit 12 months, the scores were slightly higher in the AAF-syn group than in the AAF group. Immune factors like OSM, TNSF14 and EN-RAGE were lower in the AAF-syn group than in the AAF group at baseline and visit 6 months, while

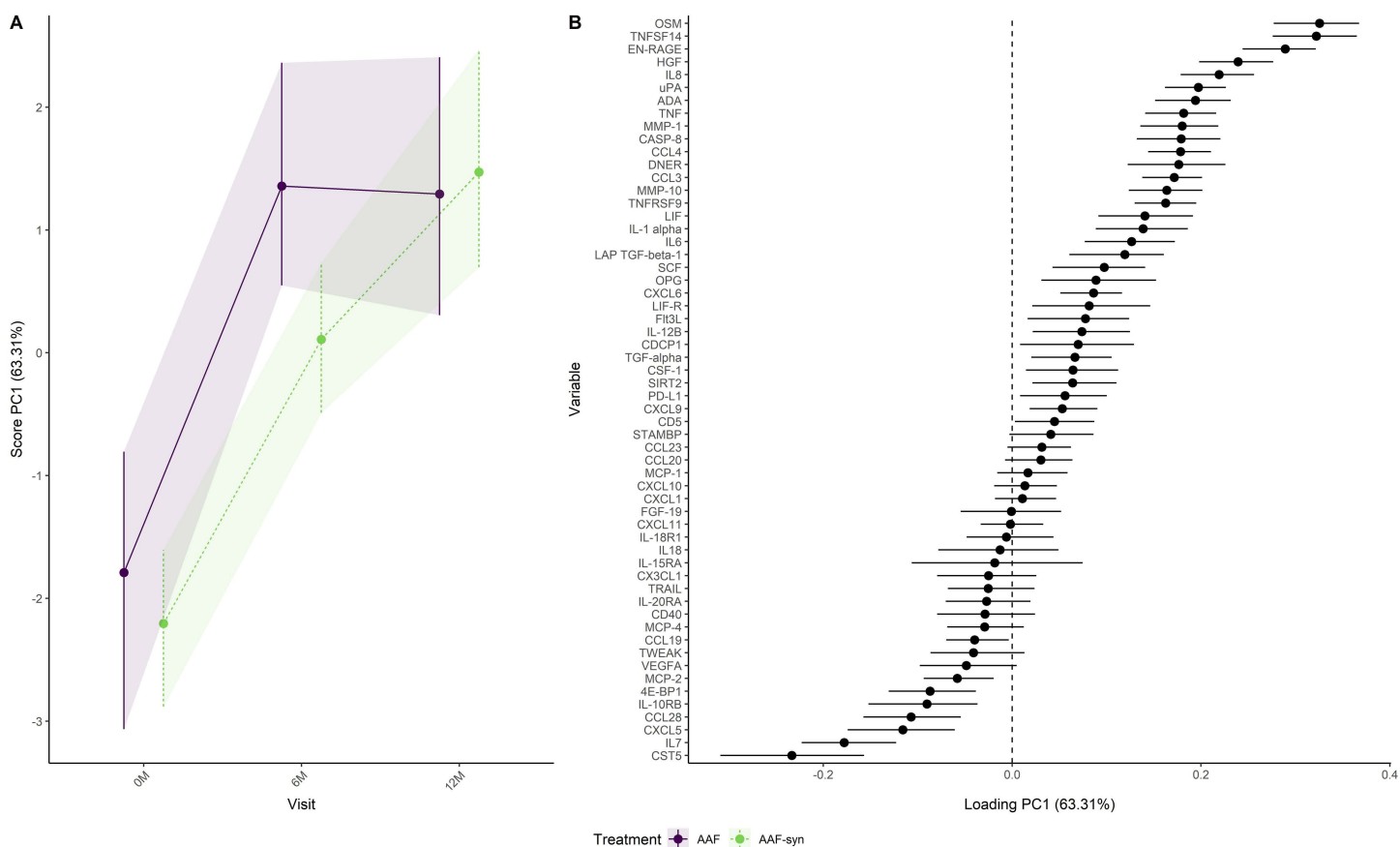

**Fig 3. Development of immune factors over visits in the AAF and AAF-syn treatment group as (A) scores and (B) loadings.** A positive loading means that the NPX value of the immune factor increases when the scores increase. A negative loading means that the NPX value of the immune factor decreases when the scores increase.

being slightly higher in the AAF-syn group than in the AAF group at visit 12 months (Fig 3B). Immune factors like CST5, IL7 and CXCL5 were higher in the AAF-syn group than in the AAF group at baseline and visit 6M, while being slightly lower in the AAF-syn group than in the AAF group at visit 12 months (Fig 3B).

Fig 4A adjusts for the trends over visits in the AAF group by using this group as a reference. The figure shows how the immune factors in the AAF-syn group differed from the AAF group. At baseline and visit 12 months, mean PC1 scores are slightly higher in the AAF-syn group than in the AAF group, but their confidence intervals showed large overlap. This indicates that immune factor levels are similar for the two treatment groups at baseline and visit 12 months. At visit 6 months, the PC1 score for the AAF-syn group was lower than for the AAF group. After adjusting for trends over visits in the AAF-syn group, infants with resolved CMA showed lower NPX values at visit 6 months for immune factors like TNF, CXCL6 and 4E-BP1, while showing higher NPX values for immune factors like IL-15RA, CST5 and CSF1.

### Significant differences between visits within treatment groups

S8 Table shows the results of the LMM analysis and subsequent post-hoc test to determine significant differences between visits within treatment groups. For OSM, EN-RAGE, TNFSF14, CST5, DNER, IL-8 and HGF similar changes were found between visits in both treatment groups (see Table 2, S8 Table and boxplots in S4 Fig in S1 File). In AAF, CASP-8 increased

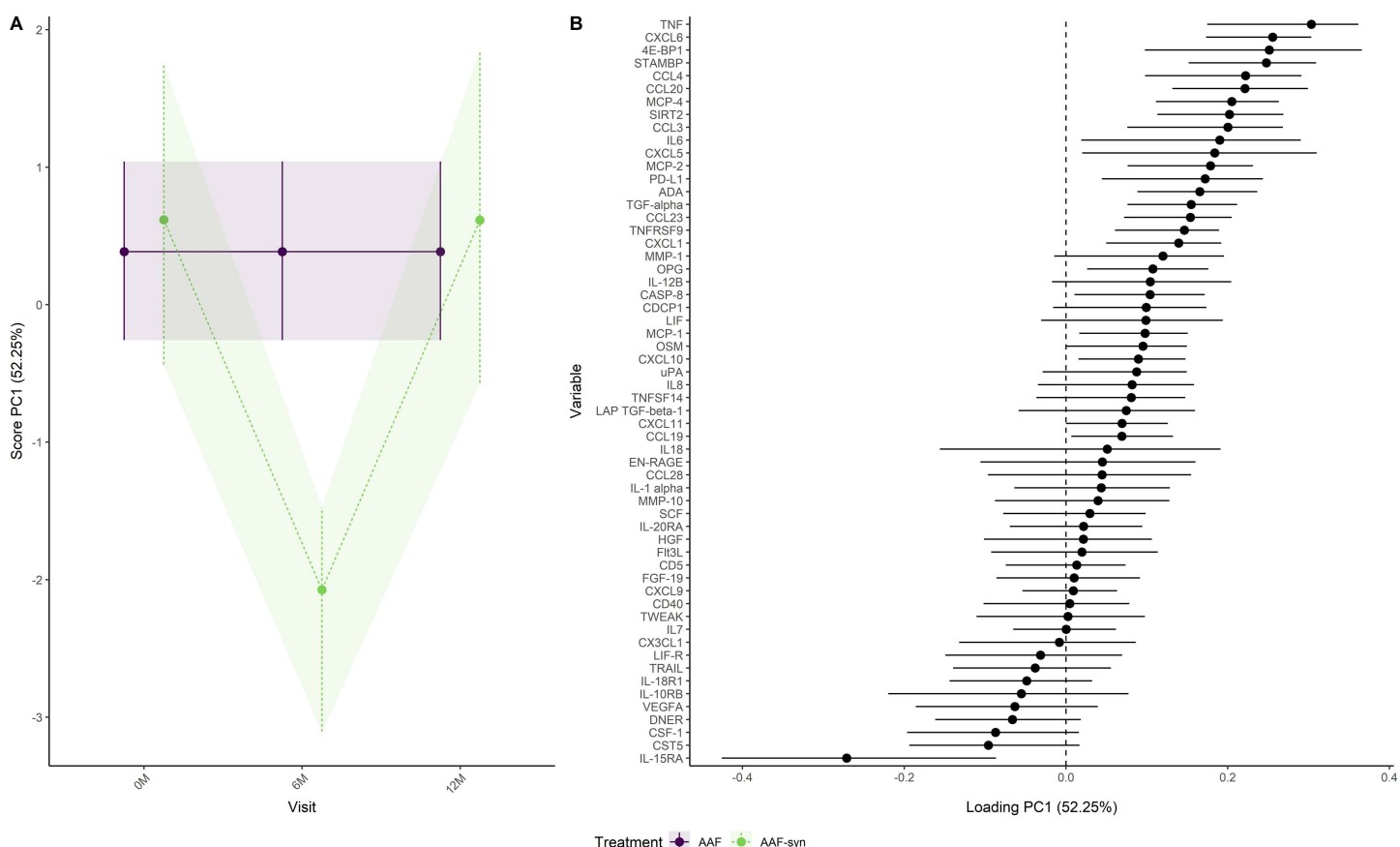

**Fig 4. Development of immune factors over visits in the AAF-syn treatment group as (A) scores and (B) loadings when using the AAF treatment group as a reference.** A positive loading means that the NPX value of the immune factor increases when the scores increase. A negative loading means that the NPX value of the immune factor decreases when the scores increase.

significantly between 0 and 6 months, while at 12 months the CASP-8 level was not significantly different from the baseline level anymore (Tables 2, S8 Table, S4 Fig in S1 File). In contrast, in AAF-syn, there was no significant difference in CASP-8 between 0 and 6 months, but CASP-8 levels were significantly higher at 12 months compared to baseline (Tables 2, S8 Table, S4 Fig in S1 File). In AAF, IL-7 did not change significantly over visits (Tables 2, S8 Table, S4 Fig in S1 File). In AAF-syn, IL-7 levels were significantly higher at 6 and 12 months compared to baseline (Tables 2, S8 Table, S4 Fig in S1 File). ADA,

**Table 2. Proteins with at least one significant difference (adjusted p-value ≤ 0.05) between visits within treatment groups as determined by linear mixed models and post hoc analysis.**

| Protein | Standard amino acid formula (AAF) Number of samples: 0M: n = 16; 6M: n = 16; 12M: n = 16 | | | | | |
| --- | --- | --- | --- | --- | --- | --- |
| | Estimate and 95% CI for the difference in mean NPX | | | Adjusted p-value | | |
| | 0M to 6M | 6M to 12M | 0M to 12M | 0M to 6M | 6M to 12M | 0M to 12M |
| **OSM** | **+2.13 [+0.51, + 3.75]** | +0.26 [−1.35, + 1.88] | **+2.40 [+0.78, + 4.01]** | **0.0032** | 0.9968 | **0.0006** |
| **EN-RAGE** | **+2.17 [+0.21, + 4.13]** | +0.06 [−1.90, + 2.02] | **+2.23 [+0.28, + 4.19]** | **0.0209** | 1.0000 | **0.0161** |
| **TNFSF14** | **+1.52 [+0.14, + 2.92]** | +0.43 [−0.96, + 1.82] | **+1.96 [+0.57, + 3.35]** | **0.0232** | 0.9421 | **0.0013** |
| **CST5** | **−0.51 [−1.00,-0.03]** | +0.03 [−0.45, + 0.51] | **−0.49 [−0.97,-0.01]** | **0.0290** | 1.0000 | **0.0457** |
| **CASP-8** | **+1.45 [+0.01, + 2.89]** | −0.55 [−1.99, + 0.90] | +0.90 [−0.54, + 2.35] | **0.0483** | 0.8779 | 0.4522 |
| **IL-7** | −0.39 [−1.15, + 0.38] | +0.04 [−0.72, + 0.81] | −0.34 [−1.10, + 0.42] | 0.6771 | 1.0000 | 0.7795 |
| **DNER** | +0.14 [−0.12, + 0.40] | +0.16 [−0.10, + 0.42] | **+0.30 [+0.05, + 0.56]** | 0.5954 | 0.4514 | **0.0116** |
| **IL-8** | +1.13 [−0.01, + 2.28] | +0.12 [−1.03, + 1.27] | **+1.26 [+0.11, + 2.40]** | 0.0549 | 0.9996 | **0.0241** |
| **HGF** | +0.89 [−0.31, + 2.08] | +0.41 [−0.78, + 1.61] | **+1.30 [+0.11, + 2.49]** | 0.2693 | 0.9112 | **0.0245** |
| **ADA** | +0.76 [−0.24, + 1.77] | −0.22 [−1.23, + 0.79] | +0.55 [−0.46, + 1.56] | 0.2421 | 0.9885 | 0.6091 |
| **TNFRSF9** | +0.74 [−0.10, + 1.58] | −0.18 [−1.02, + 0.66] | +0.56 [−0.28, + 1.40] | 0.1176 | 0.9983 | 0.3878 |
| **uPA** | +0.96 [−0.24, + 2.17] | +0.04 [−1.16, + 1.25] | +1.01 [−0.20, + 2.21] | 0.1914 | 1.0000 | 0.1557 |
| **IL-15RA** | +0.19 [−0.40, + 0.78] | −0.14 [−0.73, + 0.45] | +0.05 [−0.54, + 0.64] | 0.9321 | 0.9835 | 0.9998 |
| Protein | Amino acid formula with synbiotic (AAF-syn) Number of samples: 0M: n = 23, 6M: n = 23, 12M: n = 22 | | | | | |
| | Estimate and 95% CI for the difference in mean NPX | | | Adjusted p-value | | |
| | 0M to 6M | 6M to 12M | 0M to 12M | 0M to 6M | 6M to 12M | 0M to 12M |
| **OSM** | **+1.60 [+0.25, + 2.95]** | +0.85 [−0.51, + 2.22] | **+2.46 [+1.09, + 3.82]** | **0.0107** | 0.4560 | **< 0.0001** |
| **EN-RAGE** | **+1.73 [+0.10, + 3.37]** | +0.22 [−1.43, + 1.88] | **+1.95 [+0.30, + 3.61]** | **0.0314** | 0.9987 | **0.0112** |
| **TNFSF14** | **+1.47 [+0.31, + 2.63]** | +1.05 [−0.12, + 2.23] | **+2.52 [+1.35, + 3.70]** | **0.0051** | 0.1061 | **< 0.0001** |
| **CST5** | **−0.51 [−0.91,-0.11]** | −0.28 [−0.68, + 0.13] | **−0.79 [−1.20,-0.38]** | **0.0047** | 0.3674 | **< 0.0001** |
| **CASP-8** | +0.93 [−0.27, + 2.14] | +0.36 [−0.87, + 1.58] | **+1.29 [+0.07, + 2.51]** | 0.2202 | 0.9566 | **0.0322** |
| **IL-7** | **−0.67 [−1.30,-0.03]** | −0.19 [−0.84, + 0.45] | **−0.85 [−1.50,-0.22]** | **0.0337** | 0.9521 | **0.0027** |
| **DNER** | +0.18 [−0.04, + 0.39] | +0.04 [−0.18, + 0.26] | **+0.22[+0.00(1), + 0.44]** | 0.1704 | 0.9929 | **0.0483** |
| **IL-8** | +0.60 [−0.36, + 1.56] | +0.38 [−0.59, + 1.35] | **+0.98 [+0.01, + 1.95]** | 0.4470 | 0.8630 | **0.0460** |
| **HGF** | +0.89 [−0.10, + 1.89] | +0.49 [−0.52, + 1.50] | **+1.38 [+0.37, + 2.39]** | 0.1038 | 0.7210 | **0.0019** |
| **ADA** | +0.52 [−0.32, + 1.36] | +0.64 [−0.21, + 1.49] | **+1.16 [+0.31, + 2.01]** | 0.4628 | 0.2511 | **0.0020** |
| **TNFRSF9** | +0.37 [−0.33, + 1.08] | +0.41 [−0.30, + 1.12] | **+0.79 [+0.07, + 1.50]** | 0.6291 | 0.5423 | **0.0220** |
| **uPA** | +0.60 [−0.40, + 1.61] | +0.49 [−0.53, + 1.51] | **+1.09 [+0.08, + 2.11]** | 0.5005 | 0.7236 | **0.0281** |
| **IL-15RA** | +0.41 [−0.08, + 0.90] | **−0.59 [−1.08,-0.09]** | −0.17 [−0.67, + 0.33] | 0.1487 | **0.0116** | 0.9126 |

95% confidence intervals (CI) for the difference in mean NPX value and adjusted p-value (Benjamini-Hochberg). More detailed results are shown in S8 Table. Significant results are indicated in bold. +: increase; -: decrease.

Abbreviations: Sign.: significant; non-sign.: not significant; OSM: oncostatin-M; EN-RAGE: protein S100-A12; TNFSF14: tumor necrosis factor ligand superfamily member 14; CST5: cystatin D; CASP-8: caspase-8; IL7: interleukin-7; DNER: delta and notch-like epidermal growth factor-related receptor; IL8: interleukin-8; HGF: hepatocyte growth factor; ADA: adenosine deaminase; TNFRSF9: tumor necrosis factor receptor superfamily member 9; uPA: urokinase-type plasminogen activator; IL-15RA: interleukin-15 receptor subunit alpha.

tumor necrosis factor receptor superfamily member 9 (TNFRSF9) and uPA did not change significantly over visits in AAF, but significantly increased between 0 and 12 months in AAF-syn (Tables 2, S8 Table, S4 Fig in S1 File). IL-15RA did not change significantly over visits in AAF but significantly decreased between 6 and 12 months in AAF-syn (Tables 2, S8 Table, S4 Fig in S1 File). No significant differences between treatment groups were found within visits (S9 Table).

### Significantly different enriched pathways between visits within allergy groups

KEGG pathway analysis showed that cytokine-cytokine receptor interaction and TNF signaling pathways significantly changed between 6 and 12 months and between baseline and 12 months, with most of the proteins being increased while others decreased (Table 3, S10 Table). IL-17 and Toll-like receptor signaling pathways significantly changed between baseline and 12 months, with most of the proteins being increased while others decreased (Table 3, S10 Table).

S11 Table displays all KEGG pathways for the comparisons between allergy groups within visits. None of the results were significant (all q-values > 0.05).

### Significantly different enriched pathways between visits within treatment groups

Cytokine-cytokine receptor interaction and several signaling pathways (IL-17, Toll-like receptor, TNF) were higher at visits 6 and 12 months compared to baseline, but did not differ between visits 6 and 12 months in AAF (Table 4, S12 Table). In contrast, AAF-syn showed a significant increase in these pathways between 6 and 12 months, but no significant change from baseline at 6 and 12 months (Table 4, S12 Table).

### Significantly different enriched pathways between treatment groups within visits

At visit 6 months, cytokine-cytokine receptor interaction, as well as several signaling pathways (IL-17, TNF, Toll-like receptor) were lower in AAF-syn than in AAF (Table 5, S13 Table).

### No significant differences in the number of infections between groups or visits

S14 Table in S1 File shows that there are no significant differences between 6-month visit intervals within allergy and treatment groups. Also no significant differences between allergy and treatment groups within 6-month visit intervals were found.

Table 3. KEGG pathways (with at least 4 measured proteins) that have at least one significant difference (adjusted p-value (q-value) ≤ 0.05) between visits within allergy groups as determined with ConsensusPathDB.

| pathway | Persistent CMA at visit 12M Adjusted p-value (q-value) Number of samples: 0M: n=15, 6M: n=15; 12M: n=14 | | | Resolved CMA at visit 12M Adjusted p-value (q-value) Number of samples: 0M: n=24, 6M: n=24; 12M: n=24 | | |
|---|---|---|---|---|---|---|
| | 0M to 6M | 6M to 12M | 0M to 12M | 0M to 6M | 6M to 12M | 0M to 12M |
| Cytokine-cytokine receptor interaction (39) | 0.1855 | 0.6256 | 0.2399 | 0.2604 | **0.0002 34↑, 5↓** | **0.0050 28↑, 11↓** |
| TNF signaling pathway (13) | 0.1477 | 0.6599 | 0.3736 | 0.4397 | **0.0313 12↑, 1↓** | **0.0134 11↑, 2↓** |
| IL-17 signaling pathway (11) | 0.1719 | 1.0000 | 0.2865 | 0.2604 | 0.0510 | **0.0134 10↑, 1↓** |
| Toll-like receptor signaling pathway (12) | 0.1477 | 0.3979 | 0.1825 | 0.2298 | 0.0614 | **0.0199 11↑, 1↓** |

Pathways of which less than 10% of their members were measured and KEGG disease pathways were excluded. Between brackets the total number of measured proteins. In case of significant differences, it is also mentioned how many proteins increased or decreased in mean NPX value. More detailed results are presented in S10 Table. Significant results are indicated in bold.

Abbreviations: TNF: tumor necrosis factor; IL-17: interleukin-17.

## Discussion

In this study, the Olink® Target 96 Inflammation panel was applied on saliva samples from infants allergic to CM at baseline, of which some outgrew their CMA, while for others the CMA was persistent.

Saliva is a non-invasive source and easy to collect especially in young children. In addition, salivary constituents are known to reflect local and systemic diseases [21]. The disadvantage of the use of saliva is that analytes are usually present in much lower concentrations, demanding a highly sensitive method. For this reason the Olink® method we used, is very suitable.

**Table 4. KEGG pathways (with at least 4 measured proteins) that have at least one significant difference (adjusted p-value (q-value) ≤ 0.05) between visits within treatment groups as determined with ConsensusPathDB.**

| pathway | Standard amino acid formula (AAF) Adjusted p-value (q-value) Number of samples: 0M: n=16, 6M: n=16; 12M: n=16 | | | Amino acid formula with synbiotic (AAF-syn) Adjusted p-value (q-value) Number of samples: 0M: n=23, 6M: n=23; 12M: n=22 | | |
|---|---|---|---|---|---|---|
| | 0M to 6M | 6M to 12M | 0M to 12M | 0M to 6M | 6M to 12M | 0M to 12M |
| Cytokine-cytokine receptor interaction (39) | **0.0039** **29↑, 10↓** | 0.7156 | **0.0044** **28↑, 11↓** | 0.8422 | **0.0003** **33↑,6↓** | 0.1322 |
| IL-17 signaling pathway (11) | **0.0269** **10↑, 1↓** | 0.4911 | **0.0241** **10↑, 1↓** | 0.9961 | **0.0090** **11↑** | 0.1789 |
| Toll-like receptor signaling pathway (12) | **0.0358** **11↑, 1↓** | 0.5500 | **0.0256** **11↑, 1↓** | 0.8422 | **0.0154** **12↑** | 0.0913 |
| TNF signaling pathway (13) | **0.0462** **10↑, 3↓** | 0.5500 | **0.0241** **11↑, 2↓** | 0.9226 | **0.0154** **11↑, 2↓** | 0.2442 |

Pathways of which less than 10% of their members were measured and KEGG disease pathways were excluded. Between brackets the total number of measured proteins. In case of significant differences, it is also mentioned how many proteins increased or decreased in mean NPX value. More detailed results are presented in S12 Table. Significant results are indicated in bold.

Abbreviations: IL-17: interleukin-17; TNF: tumor necrosis factor.

**Table 5. KEGG pathways (with at least 4 measured proteins) that have at least one significant difference (adjusted p-value (q-value) ≤ 0.05) between treatment groups within visits as determined with ConsensusPathDB.**

| pathway | 0M Number of samples: AAF: 16, AAF-syn: 23 | 6M Number of samples: AAF: 16, AAF-syn: 23 | 12M Number of samples: AAF: 16, AAF-syn: 22 |
|---|---|---|---|
| | AAF-syn vs AAF Adjusted p-value (q-value) | AAF-syn vs AAF Adjusted p-value (q-value) | AAF-syn vs AAF Adjusted p-value (q-value) |
| Cytokine-cytokine receptor interaction (39) | 0.3683 | **0.0070** **11↑, 28↓** | 0.9874 |
| IL-17 signaling pathway (11) | 0.3683 | **0.0134** **11↓** | 0.9230 |
| TNF signaling pathway (13) | 0.7639 | **0.0364** **11↓, 2↑** | 0.9230 |
| Toll-like receptor signaling pathway (12) | 0.2355 | **0.0428** **10↓, 2↑** | 0.8594 |

Pathways of which less than 10% of their members were measured and KEGG disease pathways were excluded. Between brackets the total number of measured proteins. In case of significant differences, it is also mentioned how many proteins increased or decreased in mean NPX value. More detailed results are presented in S13 Table. Significant results are indicated in bold.

Abbreviations: IL-17: interleukin-17; TNF: tumor necrosis factor.

During the study, infants received AAF or AAF-syn. LMM and KEGG pathway enrichment analysis were applied to identify changes between visits within allergy (persistent CMA versus resolved CMA) or within treatment (AAF versus AAF-syn) groups.

In infants with persistent CMA, we found a significant increase of IL-15RA between baseline and 6 months, followed by a significant decrease between 6 and 12 months. IL-15RA binds interleukin 15 (IL-15) [22], and could thereby increase T-cell activity [23]. IL-15 has been reported to have anti-inflammatory effects in mice exposed to airway allergens [24]. This suggests that the decrease in IL-15RA between 6 and 12 months could reduce IL-15 levels, which in its turn could enhance the allergic response.

A significant increase in TNFSF14 and EN-RAGE (S100A12) was observed earlier in resolved CMA than in persistent CMA. Remarkably, previous studies have found that TNFα, a cytokine related to TNFSF14, and TNFRSF25, a TNFSF14 receptor, might enhance allergic inflammation [25,26]. Moreover, TNFSF14 can induce T cell activation and proliferation [27]. Furthermore, S100A12 has been reported to induce mast cell activation [28] and enhance allergic inflammation and asthma [28]. While these findings suggest a pro-inflammatory role for TNFSF14 and EN-RAGE, our results suggest that these markers instead seem to protect against allergic inflammation and induce outgrowth of CMA.

Remarkably, while IL-1alpha has been reported as linked to T cell activation [29] and being pro-inflammatory [30], we showed a significant increase for IL-1alpha between baseline and 6 months only in resolved CMA. Moreover, Willart *et al.* showed that IL-1R knockout mice could not mount a Th2 response to house dust mite [31].

In persistent CMA, IL-7 decreased significantly between baseline and 12 months. A decrease in IL-7 has been related to increased severity of atopic dermatitis in mice through up-regulation of the Th2 immune response [32].

Our finding that CASP-8 significantly increases between baseline and 12 months in persistent CMA contradicts with a previous finding in a study on allergic contact dermatitis, where reduced CASP-8 was related to increased severity of allergy [33]. In contrast, Teh *et al.* reported that CASP-8 reduces the number of Tregs [34], which are known to inhibit allergy development [35].

Our results suggest an increase in uPA, DNER, IL-8 and ADA between baseline and 12 months in resolved CMA. The increase in uPA corresponds with a recent study on outgrowth of CMA in peripheral blood mononuclear cells which also used the Olink® Target 96 Inflammation panel [36]. Furthermore, uPA has been reported as a marker associated with oesophageal allergic responses to food [37].

DNER is involved in the regulation of interferon-gamma (IFN-γ) [38], a suppressor of allergic response [39]. To the best of our knowledge, a direct relationship between DNER and allergy has not been reported earlier. Remarkably, in contrast to our results, elevated levels of IL-8 have been related to mast cell activation [40] and more severe allergy in asthma [41]. ADA degrades adenosine to inosine, thereby inhibiting T cell activity [42]. It has been suggested that elevated levels of adenosine in ADA-deficient mice are related to allergic airway inflammation [43–45]. The potential role of ADA in asthma has also been demonstrated in humans [46].

We observed that the majority of the proteins belonging to the TNF, toll-like receptor (TLR) and IL-17 signaling pathways significantly increased between 0 and 12 months in infants who have resolved CMA, but not in those with persistent CMA. TNF signaling can both induce and protect against allergic inflammation [47]. Our results suggest that TNF signaling contributes to resolution of CMA by protecting against allergic inflammation. Also, the TLR signaling pathway has been reported to be protective against inflammation [48].

Remarkably, IL-17 signaling, reported as a pathway involved in allergic inflammation [49] significantly increased over visits in the group that has resolved CMA. However, for IL-17 signaling, 9 of the 11 measured proteins overlap with TNF signaling (S10 Table). Therefore, the overall increase in the members of the IL-17 pathway can be explained by the large overlap with the TNF signaling.

Based on our results we hypothesize the following mechanisms for persistence and resolution of IgE-mediated CMA (Fig 5). In persistent CMA, an increase of IL-15RA, which binds IL-15, possibly results in increased T cell activity. The

increase of CASP-8 and decrease in IL-7 possibly reduce the number of Tregs and enhances the Th2 immune response. This could lead to the increased production of IgE antibodies, which bind to mast cells. This could result in increased degranulation of mast cells and release of inflammatory mediators after re-exposure to CM antigens. In resolved CMA, the Th2 response is possibly disrupted by an increase in DNER, which facilitates the IFN-γ production by T cells. Tregs generate adenosine [50], and the increase in ADA suggests increased degradation of the adenosine produced by Tregs to inosine, which can further inhibit T cell activity and allergic response.

A significant increase in CASP-8 was observed earlier in AAF than in AAF-syn. In AAF-syn, a significant decrease in IL-7 between baseline and the later visits was found. Increased CASP-8 and decreased IL-7 were both reported to reduce the severity of allergy (see above). This suggests that intake of AAF or AAF-syn could diminish allergy severity.

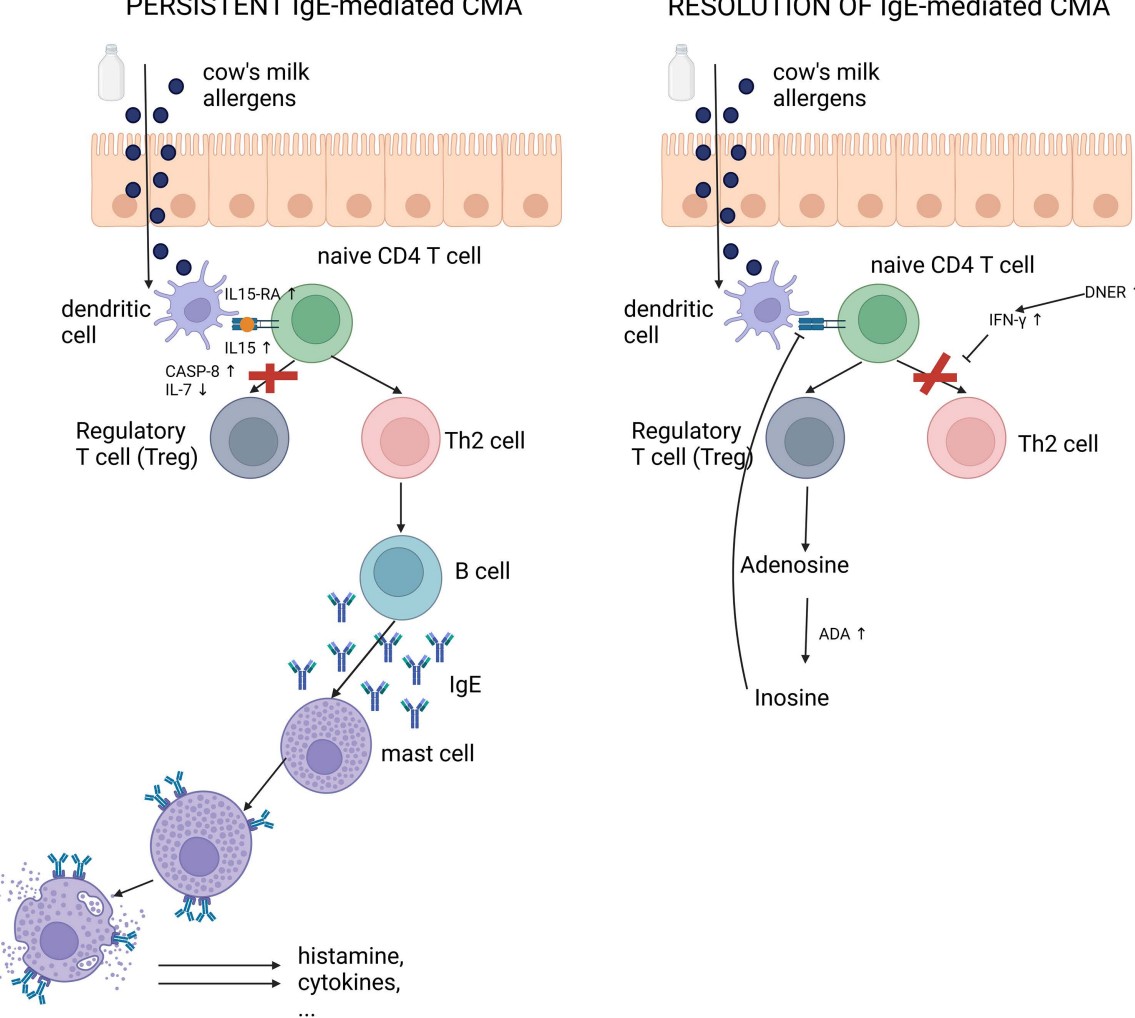

**Fig 5. Potential mechanisms for persistent CMA (left) and resolution of CMA (right) revealed by our study.** In infants with persistent CMA an increase of IL-15RA, which binds IL-15, possibly results in increased T cell activity. An increase of CASP-8 and a decrease of IL-7 could reduce the number of regulatory T cells (Tregs), and enhance the Th2 immune response, leading to the production of IgE antibodies which bind to mast cells. This could result in increased mast cell degranulation and release of inflammatory mediators after re-exposure to CM antigens. In resolved CMA, the Th2 response is possibly disrupted by an increase in DNER, which facilitates the IFN-γ production by T cells. Tregs generate adenosine which can further inhibit T cell activity and allergic response. Created with BioRender.com. Hendrickx, D. (2024) https://biorender.com/n96p662.

Our results suggest a significant increase of ADA, TNFRSF9 and uPA between baseline and 12 months in AAF-syn. This is in contradiction with previous studies suggesting that synbiotics reduce levels of TNFα, a cytokine related to TNFRSF9 [51,52]. Increases of ADA, TNFRSF9 and uPA have been associated with resolution of CMA [36]. To the best of our knowledge, a relationship between synbiotics and levels of CASP-8, IL-7, ADA and uPA has not been reported earlier.

Furthermore, a significant decrease in IL-15RA, which was related to enhanced allergic response (see above) was observed between 6 and 12 months in AAF-syn, but not in AAF.

The results mentioned above suggests that the synbiotic blend contributes to both pro- and anti- inflammatory immune response. This could probably explain why no significant effect of the synbiotic blend on resolution of CMA could be demonstrated by Chatchatee et al [11].

At visit 6 months, the majority of the proteins of the IL-17, TNF and TLR signaling pathways were lower in AAF-syn than in AAF. In the group receiving the AAF, a significant increase in the proteins of these signaling pathways could already be observed between baseline and 6 months, while in the group receiving the AAF-syn, the increase was observed later (between 6 and 12 months). Lower levels of TNF-α after intake of synbiotics have been reported earlier [51]. To the best of our knowledge, no relationship between IL-17 signaling has been demonstrated. Similar as in the comparison to allergy groups, the relationship between synbiotics and IL-17 signaling can probably be explained by the large overlap with TNF signaling. TLRs may have a role in immunotolerance mediated by probiotics [53].

Taken together, the results for the food intervention suggest the following mechanism for the synbiotic blend. A decrease in IL-7 and an increase of ADA, TNFRSF9 and uPA after intake of the synbiotic could have a positive effect on the anti-inflammatory immune response. However, this positive effect is possibly compensated by a decrease of IL-15RA, known to enhance the allergic response.

The current study has some limitations that should be mentioned. First, as all infants were allergic at baseline and therefore were not exposed to CM proteins except for the CM challenge, it is difficult to relate the effects suggested by the results to reduction of antigen presentation, therefore our hypotheses have to be taken with caution. Second, the immune system consists of a large number of interactions between cells and immune factors, which together determine immune fitness. As a consequence, an association between a single salivary immune marker and immune fitness will be moderate at best [54]. Also, we realize that in our study we did not detect previously described cytokines [8–10] such as IL-4, IL-5, IL-10, IL-3 and IFN-γ (they were below LOD in more than 20% of the samples) except for IL-10RB, a subunit of IL-10, but this marker did not show any significant results. However, studies describing for instance an increased level of IL-4 in CMA patients compared to controls [8], significantly higher levels of IL-13 in CMA compared to controls [9] or high IL-13 production and low IFN-γ production in children with CMA compared to healthy controls [10], all measured the cytokines in peripheral blood instead of saliva, which is a totally different matrix. Third, no significant differences were observed in biomarker NPX values when comparing children that outgrew their CMA to those with persistent CMA within visits. This may be caused by the small sample size. Future studies should include a larger sample size. Fourth, several biomarkers could not be (reliably) detected in saliva, and preferably, the biomarker assessments should also be conducted in blood [55]. However, for this study, there was limited availability of serum samples, and serum samples were not collected at visit 6 months. Saliva is a body fluid containing proteins, cytokines and immunoglobulins, reflecting the oral/mucosal immune response. There are recent studies showing an association between cytokines found in saliva and blood plasma [56]. Another study has compared the inflammatory profile for saliva and serum samples of IBD patients using Olink® and actually showed that approximately 40% of the markers were expressed in both saliva and serum samples, however the association between serum and saliva was poor. In addition in this study it was shown that the saliva results were of additive value for monitoring disease activity [55].

Fifth, the study was restricted to infants with CMA that received an AAF (with or without synbiotics). The PRESTO study aimed to include infants for whom AAF is recommended. Infants were frequently on a CM elimination diet before enrollment and consequently had mild but persistent symptoms [11]. No difference in outgrowth of CMA between AAF and

AAF-syn was observed in the parent study on PRESTO, and for both AAF and AAF-syn groups tolerance development was in line with natural outgrowth [11]. We observed some differences between AAF and AAF-syn at 6 months (Table 5). The DBPCFC to determine outgrowth of CMA was performed at 12 months. These findings suggest that there may have been differences due to the treatment, although they do not provide evidence of a relationship with improved outgrowth of CMA.

Future studies should assess if similar results are obtained in infants with CMA that receive other types or formula, like soy-based and extensively hydrolyzed formula.

The selection of only a subset of patients based on sample and data availability may have introduced selection bias, as the remaining participants may not be representative of the initial study population. Moreover, the limited sample size of our study may limit the generalizability of our findings. Therefore, the results should be considered as hypothesis-generating and need confirmation in future studies.

## Conclusion

In summary, infants who resolved their CMA were characterized by increased levels of IL-7, DNER and ADA over visits resulting in a reduced allergic response to CM. In contrast, in infants with persistent CMA, immune factors associated with accelerated response to CM allergens, mainly IL-15RA and CASP8, increased over visits and IL-7 decreased. KEGG pathway analysis suggests mainly the TNF signaling pathway to be important in the resolution of CM allergy. Our results show that analyzing data using the Olink® Target 96 Inflammation panel on saliva samples can reveal potential promising immunological markers and mechanisms associated with the resolution or persistence of CMA to be followed up by future research.

## Supporting information

**S1 File. Supplementary methods and results, S1-S4 Figs, S1-S5 Table and S14 Table.**
(DOCX)

**S2 File. PLOS' questionnaire on inclusivity in global research.** A complete copy of PLOS' questionnaire on inclusivity in global research.
(PDF)

**S6 Table. Results of comparing visits within allergy groups using linear mixed models and post hoc analysis.**
Allergic = persistent CMA at visit 12M; tolerant = resolved CMA at visit 12M. Number of samples: persistent CMA – 0M: 15, persistent CMA – 6M: 15, persistent CMA – 12M: 14, resolved CMA – OM: 24, resolved CMA – 6M: 24, resolved CMA – 12M: 24.
(XLSX)

**S7 Table. Results of comparing allergy groups within visits using linear mixed models and post hoc analysis.**
Allergic = persistent CMA at visit 12M; tolerant = resolved CMA at visit 12M. Number of samples: persistent CMA – 0M: 15, persistent CMA – 6M: 15, persistent CMA – 12M: 14, resolved CMA – OM: 24, resolved CMA – 6M: 24, resolved CMA – 12M: 24.
(XLSX)

**S8 Table. Results of comparing visits within treatment groups using linear mixed models and post hoc analysis.**
AAF: standard amino acid formula; AAF-syn: amino acid formula with synbiotic blend. Number of samples: AAF – OM: 16, AAF – 6M: 16, AAF-12M: 16, AAF-syn – 0M: 23, AAF-syn – 6M: 23, AAF-syn – 12M: 22.
(XLSX)

**S9 Table. Results of comparing treatment groups within visits using linear mixed models and post hoc analysis.** AAF: standard amino acid formula; AAF-syn: amino acid formula with synbiotic blend. Number of samples: AAF – OM: 16, AAF – 6M: 16, AAF-12M: 16, AAF-syn – 0M: 23, AAF-syn – 6M: 23, AAF-syn – 12M: 22. (XLSX)

**S10 Table. Results of comparing visits within allergy groups using KEGG pathway enrichment in ConsensusPathDB.** Pathways with at least 4 measured proteins are displayed. Allergic = persistent CMA at visit 12M; tolerant = resolved CMA at visit 12M. Number of samples: persistent CMA – 0M: 15, persistent CMA – 6M: 15, persistent CMA – 12M: 14, resolved CMA – OM: 24, resolved CMA – 6M: 24, resolved CMA – 12M: 24. (XLSX)

**S11 Table. Results of comparing allergy groups within visits using KEGG pathway enrichment in ConsensusPathDB.** Pathways with at least 4 measured proteins are displayed. Allergic = persistent CMA at visit 12M; tolerant = resolved CMA at visit 12M. Number of samples: persistent CMA – 0M: 15, persistent CMA – 6M: 15, persistent CMA – 12M: 14, resolved CMA – OM: 24, resolved CMA – 6M: 24, resolved CMA – 12M: 24. (XLSX)

**S12 Table. Results of comparing visits within treatment groups using KEGG pathway enrichment in ConsensusPathDB.** Pathways with at least 4 measured proteins are displayed. AAF: standard amino acid formula; AAF-syn: amino acid formula with synbiotic blend. Number of samples: AAF – OM: 16, AAF – 6M: 16, AAF-12M: 16, AAF-syn – 0M: 23, AAF-syn – 6M: 23, AAF-syn – 12M: 22. (XLSX)

**S13 Table. Results of comparing treatment groups within visits using KEGG pathway enrichment in ConsensusPathDB.** Pathways with at least 4 measured proteins are displayed. AAF: standard amino acid formula; AAF-syn: amino acid formula with synbiotic blend. Number of samples: AAF – OM: 16, AAF – 6M: 16, AAF-12M: 16, AAF-syn – 0M: 23, AAF-syn – 6M: 23, AAF-syn – 12M: 22. (XLSX)

## Acknowledgments

We acknowledge Liesbeth van Emst (Radboud UMC, Nijmegen, The Netherlands) for generating and pre-processing the Olink® data.

Members of the PRESTO study team are P. Chatchatee (Chulalongkorn University, Bangkok, Thailand); A. Nowak-Wegrzyn (New York University Langone Health, US & University of Warmia and Mazury, Olsztyn, Poland); L. Lange (St Marien Hospital, Bonn, Germany); S. Benjaponpitak (Mahidol University, Bangkok, Thailand); K. Wee Chong (KK Women's & Children's Hospital, Singapore); P. Sangsupawanich (Prince of Songkla University, Hat Yai, Thailand); M.T.J. van Ampting & M.M. Oude Nijhuis & L.F. Harthoorn & J.E. Langford, (Danone Nutricia Research, Utrecht, the Netherlands); J. Knol (Laboratory of Microbiology, Wageningen University, the Netherlands & Danone Nutricia Research, Utrecht, the Netherlands); K. Knipping (Danone Nutricia Research, Utrecht, the Netherlands); J. Garssen (Danone Nutricia Research, Utrecht, the Netherlands & Utrecht Institute for Pharmaceutical Sciences, Utrecht University, The Netherlands); V. Trendelenburg (Charité Universitätsmedizin Berlin, Germany); R. Pesek (Arkansas Children's Hospital, Little Rock, US); C.M. Davis (Texas Children's Hospital, Houston, US); A. Muraro (Padua University Hospital, Italy); M. Erlewyn-Lajeunesse (University Hospitals Southampton, Southampton, UK); A.T. Fox (Guy's and St Thomas' NHS Foundation Trust, London, UK); L.J. Michaelis (Great North Children's Hospital, Newcastle Upon Tyne Hospitals NHS Foundation Trust, UK); K. Beyer (Charité Universitätsmedizin Berlin, Germany); L. Noimark (Barts/Royal London Hospital, UK); G. Stiefel (Leicester Royal Infirmary, Leicester, UK); U. Schauer & E. Hamelmann (Ruhr-Universitat Bochem im St Josef-Hospital, Bochum, Germany); D. Peroni & A. Boner (University Hospital Verona, Italy).

## Author contributions

**Conceptualization:** Clara Belzer.

**Formal analysis:** Diana M. Hendrickx, Mengyichen Long.

**Funding acquisition:** Clara Belzer.

**Investigation:** Diana M. Hendrickx, Mengyichen Long.

**Project administration:** Harm Wopereis, Clara Belzer.

**Resources:** Renate G. van der Molen.

**Supervision:** Diana M. Hendrickx, Renate G. van der Molen, Clara Belzer.

**Writing – original draft:** Diana M. Hendrickx.

**Writing – review & editing:** Harm Wopereis, Renate G. van der Molen, Clara Belzer.

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
