## [Decision Letter · Decision Letter 0]

21 Apr 2025

PONE-D-24-53838Identification of potential inflammation markers for outgrowth of cow’s milk allergyPLOS ONE

Dear Dr. Clara Belzer

Thank you for submitting your manuscript to PLOS ONE. After careful consideration, we feel that it has merit but does not fully meet PLOS ONE’s publication criteria as it currently stands. Therefore, we invite you to submit a revised version of the manuscript that addresses the points raised during the review process.

We look forward to receiving your revised manuscript.

Kind regards,

Ahmad Salimi

Academic Editor

PLOS ONE

Journal Requirements:

2. Please include a complete copy of PLOS’ questionnaire on inclusivity in global research in your revised manuscript. Our policy for research in this area aims to improve transparency in the reporting of research performed outside of researchers’ own country or community. The policy applies to researchers who have travelled to a different country to conduct research, research with Indigenous populations or their lands, and research on cultural artefacts. The questionnaire can also be requested at the journal’s discretion for any other submissions, even if these conditions are not met.  Please find more information on the policy and a link to download a blank copy of the questionnaire here: https://journals.plos.org/plosone/s/best-practices-in-research-reporting. Please upload a completed version of your questionnaire as Supporting Information when you resubmit your manuscript

“CB, RGvdM and DMH were funded by the Dutch Research Council (NWO) and Danone Nutricia Research (grant no.16490, received by CB). The project was part of the partnership programme between NWO-TTW and Danone-Nutricia Research (https://www.nwo.nl/en/researchprogrammes/partnership/partnership-programmas/danone-nutricia-research ). The Dutch Research Council (NWO) played no role in the study design, data collection and analysis, the decision to publish and the preparation of the manuscript. Danone Nutricia Research provided the clinical data and saliva samples for this study. HW is an employee of Danone-Nutricia Research. HW reviewed the manuscript, provided comments and approved the article.”

4. In this instance it seems there may be acceptable restrictions in place that prevent the public sharing of your minimal data. However, in line with our goal of ensuring long-term data availability to all interested researchers, PLOS’ Data Policy states that authors cannot be the sole named individuals responsible for ensuring data access (http://journals.plos.org/plosone/s/data-availability#loc-acceptable-data-sharing-methods).

Before we proceed with your manuscript, please also provide non-author contact information (phone/email/hyperlink) for a data access committee, ethics committee, or other institutional body to which data requests may be sent. If no institutional body is available to respond to requests for your minimal data, please consider if there any institutional representatives who did not collaborate in the study, and are not listed as authors on the manuscript, who would be able to hold the data and respond to external requests for data access? If so, please provide their contact information (i.e., email address). Please also provide details on how you will ensure persistent or long-term data storage and availability

Reviewers' comments:

Reviewer's Responses to Questions

**Comments to the Author**

1. Is the manuscript technically sound, and do the data support the conclusions?

Reviewer #1: No

Reviewer #2: Yes

2. Has the statistical analysis been performed appropriately and rigorously? 

Reviewer #1: Yes

Reviewer #2: No

3. Have the authors made all data underlying the findings in their manuscript fully available?

Reviewer #1: Yes

Reviewer #2: Yes

4. Is the manuscript presented in an intelligible fashion and written in standard English?

Reviewer #1: Yes

Reviewer #2: Yes

5. Review Comments to the Author

Reviewer #1: I read with great interest the manuscript “Supplementary Material for: Identification of potential inflammation markers for outgrowth of cow’s milk allergy”.

The topic is of interest; however many criticisms rise regarding the study design and the inclusion criteria. My comments, as following:

-Study design: Flowchart of the subject selection process (see also S1 Fig): 169 patients were randomized to the formulas (AAF-syn 80 vs AAF 89), however only 23/80 and 17/89 respectively remained after removing patients (no feces sample available, no 16S rRNA sequencing performed, plasma immunological data incomplete). This represents a main bias for the results interpretations and conclusions

- Inclusion criteria: I am puzzled to see that more than 60% of subjects resolved their IgE allergy at 12 months of life. This is even higher than that observed in Europrevall Study (57% of those with confirmed CMA at 12 months), which was among the highest reported in literature. Most of recent powered studies evaluating the natural history of IgE mediated CMA reported a lower percentage of tolerance at 12 months of that observed in Europrevall study. This rise doubt on the criteria used to diagnose IgE cow’s milk allergy (Line 59) in the PRESTO Study

Looking at Supplementary Material demonostration of IgE mediated CMA was defined as following: “ infants sensitized to CM, confirmed by CM-specific serum IgE >0.1 kU/L and/or CM skin prick test wheal size ≥3 mm, were diagnosed with CMA as follows. For some infants, CMA diagnosis was confirmed by an open or double-blind placebo-controlled CM challenge (DBPCFC). For others, diagnosis was confirmed by a history of anaphylaxis reaction to CM reported by two physicians. Furthermore, looking at Table S1 we can see that severity of atopic dermatitis has been reported; thus it means that some infants had concomitant AD, and it is well known that AD infants may be sensitized to trofoallergens, especially CM proteins but they are not real allergic to CM. This is even more true if a confirmation OFC is not performed to confirm the diagnosis of CMA. Another point of interest regards the IgE level which was the lowest which may be considered (0.1, instead of 0.35 as often used as cut off to consider IgE sensitization). The percentage of infants who underwent a confirmatory open OFC need to be explicited as well as the percentage of infants with concomitant AD. Clinical characteristics of infants at the enrollment, especially regarding symptoms at onset, are lacking also in supplementary materials should be provided (how many patients with anaphylaxis? How many with AD and so on..)

All the above mentioned issues, regarding the inclusion criteria and the CMA diagnosis itself, joined to the small sample size, represent important methodological lacks, which call in questions the observed results and limit the validity of the conclusions.

-Last but not least, patients were fed with AAF which usually should be reserved to infants/children with more severe symptoms. Again, considering the study population, 61.5% reached tolerance at 12 months, that implies that most of patients had no severe form of CMA (if we consider “true” the diagnosis of IgE CMA and not simply sensitization).

- In the Discussion section limitations regarding the above mentioned issues are lacking (except for the small sample size)

Overall, main methodological lacks hamper the initial eagerness about the scope of the manuscript.

Reviewer #2: This study investigates the natural resolution mechanisms of cow’s milk allergy by analyzing changes in multiple cytokines using saliva, a non-invasive specimen. The finding that specific cytokine trajectories differ between children with and without resolution is noteworthy and meaningful as a hypothesis-generating study. However, the study has several limitations, including a small sample size and the retrospective nature of a secondary analysis derived from an interventional trial.

The authors are requested to address the following points. If any of these are considered limitations, they should be clearly acknowledged in the “Limitations” section:

#1 Do salivary cytokine levels reliably reflect systemic (blood) cytokine concentrations?

#2 What criteria were used to select participants from the AAF and AAF-syn interventional trial for inclusion in the present analysis?

#3 Could the dietary interventions with AAF and AAF-syn have influenced the observed resolution of CMA? It is unclear whether the cytokine changes represent responses to the intervention or natural resolution.

#4 The study mainly analyzes individual cytokine trajectories but does not include between-group comparisons or integrative analyses. Is there a reason why integrative or multivariate approaches were not applied to compare cytokine profiles across groups?

6. PLOS authors have the option to publish the peer review history of their article (what does this mean? ). If published, this will include your full peer review and any attached files.

**Do you want your identity to be public for this peer review?** For information about this choice, including consent withdrawal, please see our Privacy Policy .

Reviewer #1: No

Reviewer #2: **Yes: ** Takao Fikosawa

---

## [Author Response · Author response to Decision Letter 1]

3 Jun 2025

Reviewer #1: I read with great interest the manuscript “Supplementary Material for: Identification of potential inflammation markers for outgrowth of cow’s milk allergy”.The topic is of interest; however many criticisms rise regarding the study design and the inclusion criteria. My comments, as following:

-Study design: Flowchart of the subject selection process (see also S1 Fig): 169 patients were randomized to the formulas (AAF-syn 80 vs AAF 89), however only 23/80 and 17/89 respectively remained after removing patients (no feces sample available, no 16S rRNA sequencing performed, plasma immunological data incomplete). This represents a main bias for the results interpretations and conclusions

Authors's response: We agree that the substantial reduction in sample size due to the exclusion of these participants introduces a potential bias, which may have affected the results, interpretation and conclusions.

In the first sentence of the subsection “Sample collection and experimental design” of the methods section, we added “based on sample and data availability” (line 82).

We added the following paragraph to the discussion: “The selection of only a subset of patients based on sample and data availability may have introduced selection bias, as the remaining participants may not be representative of the initial study population. Moreover, the limited sample size of our study may limit the generalizability of our findings. Therefore, the results should be considered as hypothesis-generating and need confirmation in future studies.”(lines 489-493)

- Inclusion criteria: I am puzzled to see that more than 60% of subjects resolved their IgE allergy at 12 months of life. This is even higher than that observed in Europrevall Study (57% of those with confirmed CMA at 12 months), which was amdong the highest reported in literature. Most of recent powered studies evaluating the natural history of IgE mediated CMA reported a lower percentage of tolerance at 12 months of that observed in Europrevall study. This rise doubt on the criteria used to diagnose IgE cow’s milk allergy (Line 59) in the PRESTO Study.

Author's response: We like to clarify that the infants in our study were older than 12 months of age when they developed tolerance to CM. At baseline, they were between 3 and 13 months old (see methods section, line 129). On average, they were ca 9 months. We added the mean age at baseline in the methods (line 129). So this means that after being 12 months in the study, they were between 15 and 25 months old (mean ca 21 months). In the complete PRESTO cohort, the % of subjects that resolved their IgE allergy after being 12 months in the study was 49% and in line with natural outgrowth (see Chatchatee et al, 2022, Food allergy and gastrointestinal disease, Volume 149, Issue 2, p650-658, https://www.jacionline.org/article/S0091-6749(21)01053-8/fulltext )(see also methods section, lines 92-93). The higher percentage of infants that resolved their IgE allergy after being 12 months in the study in our subset of PRESTO is due to the selection procedure based on sample and data availability. In the discussion of the revised manuscript we indicate that the subset may not be representative of the initial study population because of the selection procedure (see answer to your previous question on study design).

Looking at Supplementary Material demonostration of IgE mediated CMA was defined as following: “ infants sensitized to CM, confirmed by CM-specific serum IgE >0.1 kU/L and/or CM skin prick test wheal size ≥3 mm, were diagnosed with CMA as follows. For some infants, CMA diagnosis was confirmed by an open or double-blind placebo-controlled CM challenge (DBPCFC). For others, diagnosis was confirmed by a history of anaphylaxis reaction to CM reported by two physicians. Furthermore, looking at Table S1 we can see that severity of atopic dermatitis has been reported; thus it means that some infants had concomitant AD, and it is well known that AD infants may be sensitized to trofoallergens, especially CM proteins but they are not real allergic to CM. This is even more true if a confirmation OFC is not performed to confirm the diagnosis of CMA. Another point of interest regards the IgE level which was the lowest which may be considered (0.1, instead of 0.35 as often used as cut off to consider IgE sensitization). The percentage of infants who underwent a confirmatory open OFC need to be explicited as well as the percentage of infants with concomitant AD. Clinical characteristics of infants at the enrollment, especially regarding symptoms at onset, are lacking also in supplementary materials should be provided (how many patients with anaphylaxis? How many with AD and so on..)All the above mentioned issues, regarding the inclusion criteria and the CMA diagnosis itself, joined to the small sample size, represent important methodological lacks, which call in questions the observed results and limit the validity of the conclusions.

Author's response: We added the clinical characteristics of infants at the enrollment to the supplementary material (see S1 Table).

The cut-off of 0.1 kU/L for the IgE level (instead of 0.35) was decided with the clinical investigators as a clinically relevant cutoff for determining sensitization, specifically for young children as included in PRESTO. In addition, this criterion was always applied in combination with an open food challenge or a DBPCFC, unless the intake of cow's milk resulted in an anaphylactic reaction. The combination was necessary for eligibility. We added the following to S1 File, section 1:”The criterion of CM-specific serum IgE >0.1 kU/L was always applied in combination with an open food challenge or a DBPCFC, unless the intake of cow's milk resulted in an anaphylactic reaction. The combination was necessary for eligibility.”

-Last but not least, patients were fed with AAF which usually should be reserved to infants/children with more severe symptoms. Again, considering the study population, 61.5% reached tolerance at 12 months, that implies that most of patients had no severe form of CMA (if we consider “true” the diagnosis of IgE CMA and not simply sensitization).

Author's response: We agree with the reviewer that symptoms were mild during enrollment in PRESTO, but that does not mean they did not have severe CMA. Multiple cow's milk elimination diets had been used frequently before enrollment. We added the following paragraph to the discussion: “The PRESTO study aimed to include infants for whom AAF is recommended. Infants were frequently on a CM elimination diet before enrollment and consequently had mild but persistent symptoms[11].”(lines 479-480)

- In the Discussion section limitations regarding the above mentioned issues are lacking (except for the small sample size)Overall, main methodological lacks hamper the initial eagerness about the scope of the manuscript.

Author's response: The limitations mentioned by the reviewer are discussed in our revised paper, see my answers to the previous questions.

Reviewer #2: This study investigates the natural resolution mechanisms of cow’s milk allergy by analyzing changes in multiple cytokines using saliva, a non-invasive specimen. The finding that specific cytokine trajectories differ between children with and without resolution is noteworthy and meaningful as a hypothesis-generating study. However, the study has several limitations, including a small sample size and the retrospective nature of a secondary analysis derived from an interventional trial. The authors are requested to address the following points. If any of these are considered limitations, they should be clearly acknowledged in the “Limitations” section:

#1 Do salivary cytokine levels reliably reflect systemic (blood) cytokine concentrations?

Author's response: We added the following paragraph to the discussion: “Saliva is a body fluid containing proteins, cytokines and immunoglobulins, reflecting the oral/mucosal immune response. There are recent studies showing an association between cytokines found in saliva and blood plasma. Another study has compared the inflammatory profile for saliva and serum samples of IBD patients using Olink and actually showed that approximately 40% of the markers were expressed in both saliva and serum samples, however the association between serum and saliva was poor. In addition in this study it was shown that the saliva results were of additive value for monitoring disease activity”.(lines 471-477)

#2 What criteria were used to select participants from the AAF and AAF-syn interventional trial for inclusion in the present analysis?

Author's response: See S1 fig in S1 file. The participants were selected based on sample and availability of other types of data in function of the final aim of our project (Earlyfit): to combine data from different omics/platforms into a predictive model. Therefore participants were selected by availability of samples and other (-omics and non-omics) data types (participants with no feces sample available, no 16S rRNA sequencing performed or plasma immunological data incomplete were removed). We agree that the substantial reduction in sample size due to the exclusion of these participants introduces a potential bias, which may have affected the results, interpretation and conclusions.

In the first sentence of the subsection “Sample collection and experimental design” of the methods section, we added “based on sample and data availability”.(line 82).

We added the following paragraph to the discussion: “The selection of only a subset of patients based on sample and data availability may have introduced selection bias, as the remaining participants may not be representative of the initial study population. Moreover, the limited sample size of our study may limit the generalizability of our findings. Therefore, the results should be considered as hypothesis-generating and need confirmation in future studies.”(lines 489-493)

#3 Could the dietary interventions with AAF and AAF-syn have influenced the observed resolution of CMA? It is unclear whether the cytokine changes represent responses to the intervention or natural resolution.

Author's response: It is unlikely that the dietary interventions with AAF and AAF-syn have influenced the observed resolution of CMA, as no difference in outgrowth of CMA between AAF and AAF-syn was observed in the parent study on PRESTO. We added the following paragraph to the discussion: “No difference in outgrowth of CMA between AAF and AAF-syn was observed in the parent study on PRESTO, and for both AAF and AAF-syn groups tolerance development was in line with natural outgrowth[11]. We observed some differences between AAF and AAF-syn at 6 months (Table 5). The DBPCFC to determine outgrowth of CMA was performed at 12 months. These findings suggest that there may have been differences due to the treatment, although they do not provide evidence of a relationship with improved outgrowth of CMA.”(lines 481-486)

#4 The study mainly analyzes individual cytokine trajectories but does not include between-group comparisons or integrative analyses. Is there a reason why integrative or multivariate approaches were not applied to compare cytokine profiles across groups?

Author's response: Thank you for suggesting to include also a multivariate analysis. In our revised manuscript we also included a multivariate analysis with Repeated Measures ANOVA–Simultaneous Component Analysis Plus (RM-ASCA+). The method is described in the subsection “Statistical analysis” of the Methods section (lines 164-169). The results are reported in the sections “Development of immune factors over visits within allergy groups” (lines 183-207) and “Development of immune factors over visits within treatment groups” in the Results section.(lines 239-267)

---

## [Decision Letter · Decision Letter 1]

18 Aug 2025

Identification of potential inflammation markers for outgrowth of cow’s milk allergy

PONE-D-24-53838R1

Dear Dr. Belzer,

We’re pleased to inform you that your manuscript has been judged scientifically suitable for publication and will be formally accepted for publication once it meets all outstanding technical requirements.

Kind regards,

Dost Muhammad Khan, PhD

Academic Editor

PLOS ONE

Additional Editor Comments (optional):

Reviewers' comments:

Reviewer's Responses to Questions

**Comments to the Author**

1. If the authors have adequately addressed your comments raised in a previous round of review and you feel that this manuscript is now acceptable for publication, you may indicate that here to bypass the “Comments to the Author” section, enter your conflict of interest statement in the “Confidential to Editor” section, and submit your "Accept" recommendation.

Reviewer #1: (No Response)

Reviewer #2: All comments have been addressed

2. Is the manuscript technically sound, and do the data support the conclusions?

Reviewer #1: Yes

Reviewer #2: Yes

3. Has the statistical analysis been performed appropriately and rigorously? 

Reviewer #1: N/A

Reviewer #2: Yes

4. Have the authors made all data underlying the findings in their manuscript fully available?

Reviewer #1: Yes

Reviewer #2: Yes

5. Is the manuscript presented in an intelligible fashion and written in standard English?

Reviewer #1: Yes

Reviewer #2: Yes

6. Review Comments to the Author

Reviewer #1: The authors have properly addressed most of my concerns, also adding further study limitations in the Discussion section, as suggested.

In the revised version of the manuscript the authors state that “Infants were frequently on a CM elimination diet before enrollment and consequently had mild but persistent symptoms[11].”(lines 479-480)”.

It is not clear what type of CM elimination diet they did before enrolment (multiple cow’s milk elimination diets, I suppose different formulas before enrolment, but it should be clarified) and what it exactly means that they consequently had mild but persistent symptoms. The last question: it is still unclear how CMA is defined “severe” if symtoms was mild; did they mean severe as persisten? If so, please add clarification to this point

Reviewer #2: The authors have thoroughly and appropriately addressed all the points raised in the review. Each comment has been carefully considered, and the corresponding revisions made to the manuscript are clear and satisfactory. I appreciate the authors’ efforts in improving the quality and clarity of the manuscript based on the feedback provided.

7. PLOS authors have the option to publish the peer review history of their article (what does this mean? ). If published, this will include your full peer review and any attached files.

**Do you want your identity to be public for this peer review?** For information about this choice, including consent withdrawal, please see our Privacy Policy .

Reviewer #1: No

Reviewer #2: **Yes: ** Takao Fujisawa, MD, PhD

---

## [Editor Report · Acceptance letter]

PONE-D-24-53838R1

PLOS ONE

Dear Dr. Belzer,

I'm pleased to inform you that your manuscript has been deemed suitable for publication in PLOS ONE. Congratulations! Your manuscript is now being handed over to our production team.

Kind regards,

on behalf of

Dr. Dost Muhammad Khan

Academic Editor

PLOS ONE